# Relationships between Physical Activity, Sleeping Time, and Psychological Distress in Community-Dwelling Elderly Japanese

**DOI:** 10.3390/medicina55070318

**Published:** 2019-06-27

**Authors:** Yukio Yamamoto, Hiromi Suzuki, Yutaka Owari, Nobuyuki Miyatake

**Affiliations:** 1Department of Hygiene, Faculty of Medicine, Kagawa University, Miki, Kagawa 761-0793, Japan; 2Department of Judo Therapy, Shikoku Medical College, Utazu, Kagawa 769-0205, Japan

**Keywords:** physical activity, psychological distress, sedentary behavior, sleeping, walking

## Abstract

*Background and objectives:* It is well-known that lifestyle is closely associated with psychological distress in many elderly subjects. However, the effect of intervention with physical activity and/or sleeping on psychological distress has not been fully discussed. The purpose of this cross-sectional study was to investigate the relationships between physical activity, sleeping time, and psychological distress in community-dwelling elderly Japanese subjects. *Materials and Methods:* A total of 108 elderly Japanese (31 men and 77 women) subjects were enrolled in this cross-sectional study. Psychological distress was evaluated using the K6 questionnaire. Physical activity, including sedentary behavior, was measured using a tri-accelerometer. Sleeping time was evaluated using a self-reported questionnaire. *Results:* The median of the K6 scores was 1.0 (0–18), and the sedentary behavior (%) and walking time (minutes/day) were 57.2 ± 10.7 and 80.7 (17.9–222.4), respectively. Sleeping time was negatively correlated with psychological distress. In addition, multiple linear regression showed that walking time and sleeping time were important factors for psychological distress, even after adjusting for confounding factors. *Conclusions:* These results suggest that increased walking time and sleeping time may be beneficial for reducing psychological distress in community-dwelling elderly Japanese subjects.

## 1. Introduction

The number of elderly people has been markedly increasing in Japan, with the proportion of elderly subjects reported to be 27.7% in a national survey [1]. Health problems among elderly subjects, especially psychological problems such as depression, have become a public health challenge in Japan. The percentage of elderly people living in rural areas who have depressive symptoms is 33.5%, which is closely associated with activities of daily living and quality of life [2]. Moreover, even in large-scale surveys, Haseda et al. pointed out the association between the inequality of community social capital and depressive symptoms among older Japanese adults [3]. Therefore, appropriate strategies for preventing and reducing psychological distress in elderly Japanese subjects are urgently required.

It is well-known that lifestyle factors, including physical activity [4,5,6,7,8,9], diet [10,11], sleeping [12,13,14,15,16,17], cigarette smoking [18,19], and alcohol consumption [20,21], are closely linked to psychological distress. The prevalence of smoking and drinking habits in elderly Japanese is lower than in younger Japanese [22]. An accurate diet survey is clinically difficult for elderly people, due to their recall difficulty.

The tri-accelerometer has been developed and used for the evaluation of physical activity and sedentary behavior in clinical practice. Although there have been many reports on the relationship between physical activity and psychological distress based on self-reported questionnaires [4,5,6,7,8,9], there are few reports which have used objective methods, such as a tri-accelerometer with high validity [23,24,25], to evaluate physical activity, including sedentary behavior, in elderly Japanese subjects. In addition, although average sleeping time has often been evaluated by use of simple questionnaires, seven-day average sleeping records would provide more accurate and beneficial information.

In this cross-sectional study, we evaluated the relationships among physical activity, sleeping time, and psychological distress in community-dwelling elderly Japanese subjects living in rural areas in western Japan.

## 2. Materials and Methods

### 2.1. Participants

A total of 108 elderly Japanese subjects (31 men and 77 women) among 130 candidates were enrolled in this cross-sectional study (Figure 1). The subjects (1) provided written informed consent; (2) voluntarily participated in a monthly health-promotion class at a college in Utazu, Kagawa, Japan from March–May 2018; and (3) underwent measurements of physical activity using a tri-accelerometer and self-reported questionnaire in the mentioned class.

Ethical approval was obtained from the Ethical Committee of Shikoku Medical College, Utazu, Japan (approval number: H30-1; 18 February 2018).

### 2.2. Clinical Parameters

Age, height (cm), body weight (kg), medications, lifestyles (i.e., smoking habits, exercise habits, and drinking habits), physical activity, sleeping time (min), and psychological distress were recorded. Body mass index (BMI) was calculated as body weight (kg)/(height (m))^2^. Medications and lifestyles were evaluated according to the Specific Health Checkups questionnaire by the Ministry of Health, Labour, and Welfare Japan [26]. Physical activity was evaluated using a tri-accelerometer (Active Style Pro HJA-750C, Omron Healthcare, Kyoto, Japan), which is one of most commonly used devices with high reliability and validity in Japan, as previously described [27,28]. As in our previous report [29], walking time (min/day), three categorized physical activities (sedentary behavior of ≤1.5 metabolic equivalents (Mets) (min/day), 1.6–2.9 Mets (min/day), and ≥3 Mets (min/day)), and the proportion of physical activity (%/day) were used for this analysis. Walking time was defined as the integrated value of the time classified as walking using a tri-accelerometer. Total sleeping time (min/day) and time in bed (min/day) were evaluated, using a self-reported questionnaire, for seven consecutive days. The average over the seven days was also used for this analysis, as previously described [30]. Psychological distress was measured using the Kessler six-item questionnaire (K6), as previously described [31]. The six questions were as follows [32,33]: “Over the last month, how often did you feel: (1) nervous, (2) hopeless, (3) restless or fidgety, (4) so sad that nothing could cheer you up, (5) that everything was an effort, or (6) worthless?” Participants were asked to respond by choosing “all of the time” (4 points), “most of the time” (3 points), “some of the time” (2 points), “a little of the time” (1 point), or “none of the time” (0 points). The possible total point scores, thus, ranged from 0 to 24. The K6 was developed using modern psychometric theory and has been demonstrated to be superior to some existing scales in brevity and psychometric properties [34,35,36]. The Japanese version of the K6 was recently developed, using the standard back-translation method, and has been validated [32]. A score of five or more points was taken to indicate a condition of psychological distress [8].

### 2.3. Statistical Analysis

Data, distributed both normally and non-normally, were evaluated by the Shapiro–Wilk test. Then, data were expressed as the mean ± standard deviation (SD) (normal distribution) and median (minimum−maximum) (non-normal distribution) and number of subjects (%). The Spearman signed rank correlation coefficient was used to evaluate the relationship between psychological distress and clinical parameters. In addition, multiple linear regression analysis was used to identify factors related to psychological distress, taking *p* < 0.05 to indicate significance. The variance inflation factor (VIF) was used to assess multi-collinearity. Statistical analysis was performed using JMP13.2 (SAS Institute Inc., Cary, NC, USA).

## 3. Results

Clinical profiles of the enrolled subjects are summarized in Table 1. The relationships between psychological distress and clinical parameters were evaluated by the Spearman rank correlation coefficient in order to evaluate the relationships among physical activity, sleeping time, and psychological distress (see Table 2). Total sleeping time was negatively correlated with psychological distress. However, the relationships between other parameters and psychological distress were not significant.

Indeed, we identified factors related to psychological distress in community-dwelling elderly Japanese subjects by multiple linear regression analysis (Table 3). We used psychological distress as the dependent variable, and sex, age (years), walking time (min/day), sedentary behavior (≤1.5 Mets (%/day)), and total sleeping time (min/day) as independent valuables, which are thought to be clinically important, as per the literature [37,38,39,40,41]. As a result, walking time (min/day) and total sleeping time (min/day) were negatively associated with psychological distress, even after adjusting for age, sex, and sedentary behavior. In addition, after further adjusting for smoking and drinking habits, the significance of walking time (standardized *β* = −0.237 and *p* = 0.040) and total sleeping time (standardized *β* = −0.245 and *p* = 0.008) were not attenuated.

## 4. Discussion

In this study, we explore the relationship between psychological distress and lifestyle in community-dwelling elderly Japanese subjects and found that walking time and sleeping time were negatively associated with psychological distress.

Okoro et al. reported, using the K6, that physical activity was significantly lower in subjects with psychological distress than in subjects without psychological distress; among 78,886 U.S. adults in a cross-sectional study [8]. Mc Dowell et al. also found that increased physical activity was associated with an improvement of depressive symptoms on the Center for Epidemiologic Studies Depression Scale (CESD) [7]. In Japan, Ishihara et al. showed that, among middle-aged men and women, there was a negative correlation between physical activity and psychological distress, as evaluated by the General Health Questionnaire-28 (GHQ-28) [5]. In a longitudinal study, exercise was significantly and negatively associated with psychological distress in 50–59 year-old Japanese subjects [9]. Physical activity in these reports was evaluated by self-reported questionnaires. In this study, we evaluated physical activity using a tri-accelerometer and found walking time to be negatively associated with psychological distress by multiple linear regression analysis. There was no significant relationship between sedentary behavior (%), evaluated using a tri-accelerometer, and psychological distress in community-dwelling elderly Japanese in another cross-sectional study [29]. The enrolled subjects were more health-conscious than the average and they had lower psychological distress. In clinical practice, increasing the walking time of subjects may reduce psychological distress.

Atkins et al. reported that a much longer sleeping time was associated with psychological distress [12]. Cunningham et al. also found that subjects with psychological distress had much longer or shorter sleeping times than subjects without psychological distress [13]. In Japan, short sleeping times have been found to significantly and adversely affect mental health, compared with normal sleeping times, in adults [15]. In a longitudinal study, shorter sleeping times were found to be closely related to psychological distress in 5000 adolescents [14]. Sleeping time is often surveyed using a self-reported questionnaire sheet, asking “On average, how many hours do you sleep in a 24 h period?”

In this study, using a seven-day self-reported questionnaire, we found a significant and negative relationship between sleeping time and psychological distress. In addition, multiple linear regression analysis revealed walking time and sleeping time to be important factors in psychological distress. The sleeping time of the subjects in this study was 399.0 ± 70.3 min/day, which is shorter than that reported for elderly Japanese [42]. Sleeping time was negatively correlated with psychological distress. Taken together, improving sleeping times might be associated with reducing psychological distress in community-dwelling Japanese subjects.

There were several potential limitations in this study. First, the study was cross-sectional in design. Second, the elderly subjects were voluntarily enrolled in health classes, and may, thus, have been more health-conscious than the general population. Third, we could not identify the mechanism of association between psychological distress and walking time, or between psychological distress and sleeping time. Fourth, participants in this survey were community-dwelling and live in rural areas, and so may not provide a good representation of elderly Japanese subjects.

Nevertheless, we found that walking time and total sleeping time were closely associated with psychological distress in community-dwelling elderly Japanese subjects. Consistent with our findings, Justino et al. showed that higher physical activity was closely associated with lower depression using the Geriatric Depression Screening scale (GDS-15) in an intervention study [43]. Blake et al. found that sleep intervention reduced sleep disorders and anxiety symptoms [44]. Therefore, increasing walking time and sleeping time might be associated with reducing psychological distress in clinical practice. Further longitudinal and large-sample studies are required.

## 5. Conclusions

Increasing walking time and sleeping time may be beneficial for reducing psychological distress in community-dwelling elderly Japanese subjects.

## Figures and Tables

**Figure 1 medicina-55-00318-f001:**
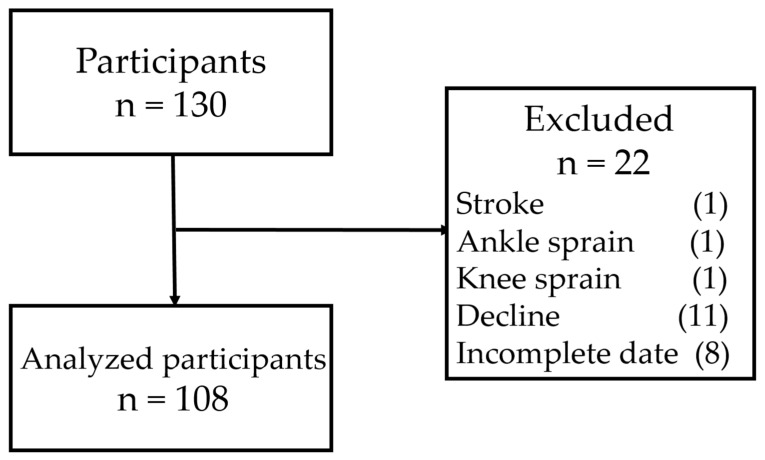
Flow diagram of analysis.

**Table 1 medicina-55-00318-t001:** Clinical characteristics of enrolled subjects.

	Total (n = 108)	
Age (years)	74.0	(65–88)
Height (cm)	154.5	(139–178)
Body weight (kg)	54.5	(36–89)
Body mass index (kg/m^2^)	22.8	±2.9
Exercise (Mets × h/w)	4.6	(0.32–13.4)
Number of Steps (steps/day)	5456.6	(525.9–20372.7)
Walking time (min/day)	80.7	(17.9–222.4)

≦1.5 Mets (min/day)	501.8	±124.0
1.6–2.9 Mets (min/day)	285.0	±78.5
≧3 Mets (min/day)	74.9	(5.7–180.8)
≦1.5 Mets (%/day)	57.2	±10.7
1.6–2.9 Met (%/day)	33.1	±8.4
≧3 Mets (%/day)	9.0	(0.61–23.28)
Total sleeping time (min/day)	399.0	±70.3
Time in bed (min/day)	480.5	(359–820)
K6 score	1.0	(0–18)
Psychological distress, n (%)	18 (16.7)	
Medication		
Hypertension, n (%)	44 (40.7)	
Diabetes Mellitus, n (%)	7 (6.5)	
Dyslipidemia, n (%)	29 (26.9)	
Medical history		
Stroke, n (%)	5 (4.6)	
Heart disease, n (%)	12 (11.1)	
Chronic kidney disease, n (%)	1 (0.9)	
Anemia, n (%)	16 (14.8)	
Smoking habits, n (%)	2 (1.9)	
Exercise habits, n (%)	48 (44.4)	
Drinking habits, n (%)	28 (25.9)	

Results are number of subjects (%) and mean ± standard deviation or median (minimum–maximum); Mets: Metabolic equivalents.

**Table 2 medicina-55-00318-t002:** Spearman rank correlation coefficient (rs) between psychological distress and clinical parameters.

	rs	p
Sex	0.075	0.462
Age (years)	0.041	0.671
Body mass index (kg/m^2^)	–0.018	0.854
Exercise (Mets/h/w)	–0.021	0.826
Number of Steps (steps/day)	–0.084	0.387
Walking time (min/day)	–0.095	0.326
≦1.5 Mets (min/day)	–0.042	0.667
≧3 Mets (min/day)	–0.016	0.867
≦1.5 Mets (%/day)	–0.019	0.847
1.6–2.9 Mets (%/day)	0.054	0.579
≧3 Mets (%/day)	0.006	0.952
Total sleeping time (min/day)	–0.224	**0.020**
Time in bed (min/day)	–0.078	0.424

Spearman rank correlation coefficient; Bold values express statistical significance (*p* < 0.05); Mets: Metabolic equivalents.

**Table 3 medicina-55-00318-t003:** Multiple linear regression analysis to identify the association between psychological distress and daily activities.

Objective Variable	Explanatory Variables	b	95% Cl			Standardized *β*	*p*	VIF
Psychological distress	Constant	16.214	3.053	to	29.376			
	Sex	−0.055	−1.513	to	1.734	0.013	0.893	1.125
	Age (years)	−0.064	−0.211	to	0.074	−0.095	0.341	1.126
	Walking time (min/day)	−0.026	−0.048	to	−0.004	−0.276	**0.019**	1.548
	≦1.5 Mets (%/day)	−2.373	−10.278	to	5.533	−0.068	0.553	1.482
	Total sleeping time (min/day)	−0.012	−0.022	to	−0.002	−0.231	**0.016**	1.024

Bold values express statistical significance (*p* < 0.05); Cl: confidence intervals.

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
