# Peer review of "Relationships between Physical Activity, Sleeping Time, and Psychological Distress in Community-Dwelling Elderly Japanese"

_medicina, 2019, doi:10.3390/medicina55070318_

Round 1
Reviewer 1 Report
Although the topic is interesting, I believe the authors failed to show the contributions of the paper to the existing literature. Such contribution should be stressed in a revised version of the manuscript.
I have additional suggestions regarding specific sections of the paper.
Title:
· The term “effects” should not be used, considering that this was a cross-sectional and correlational study, and it is not possible to establish causality.
· Regarding Materials and Methods, measures should be presented in the same order that the variables were mentioned in the study goal.
Abstract:
· The last sentence in Background and objectives includes information pertaining to Materials and Methods (e.g., study design).
Introduction:
· In the first paragraph, prevalence data (%) should be presented for dementia and depression.
· In lines 36-38, the authors should explain the focus on only two of the several lifestyle variables that are relevant to psychological distress.
· In lines 41-42, the authors should highlight the advantages of using objective methods to assess the study variables.
· In the last sentence in line 43, the authors highlight a limitation that exists in the current study. If the current study in any way helps to minimize that limitation, this should be stressed here.
· The authors should explain whether community-dwelling elderly Japanese individuals are representative of elderly Japanese individuals regarding the study variables. If there are relevant specificities, these should be noted here.
Materials and methods
· The authors should explain which criteria led to 130 subjects being eligible to the study. The authors should also explain how the 108 participants were chosen from the previous group.
· In line 50, the authors should provide further information regarding the health promotion class. Was this a one-time event or were there multiple sessions? Were the participants assessed before or after the class?
· In line 69, more information should be provided regarding K6, namely number of items, response scale, scoring, and internal consistency.
· In lines 84-91, the authors should not duplicate information from the table.
· Gender comparisons presented in Table 1 are not in accordance with the study goal and no background is provided in the Introduction.
· In line 96, the authors should explain the criteria for selecting the independent variables for the regression analysis.
· In Table 1, a significant gender difference regarding total sleeping time was not highlighted.
· The authors should explain why the correlations presented in Table 2 were calculated separately for each gender. They should also explain the criteria for selecting the correlates, as not all the variables in Table 1 were considered (e.g., smoking habits, drinking habits).
· In Table 2, the authors should provide the test results for all the analyses. I find it insufficient to only present the significance values.
· Throughout the text, “psychological distress” should be used instead of “K6 scores”, as the focus should be on the variable, not the measure.
· Contrary to previous analyses, in Table 3 there was not a separation according to gender. The authors should explain this option.
Discussion
· Whenever a significant association is mentioned, it should be stated whether it is positive or negative (e.g., lines 121 and 134).
· In lines 125-127, the authors should try to explain this result, as it is inconsistent with most literature.
· In lines 127 and 141, the word “Therefore” should not be used, as the subsequent conclusion does not follow from the previous sentence.
· Considering the limitation presented in lines 146-148, clinical implications like those in lines 157-158 may not be justified.
Reviewer 2 Report
The aim of the manuscript “Effects of Physical Activity and Sleeping on Psychological Distress in Community-Dwelling Elderly Japanese Subjects” is to investigate the effect of physical activity and sleeping on psychological distress in community dwelling elderly Japanese subject. The topic is interesting and an important contribution of the work is the evaluation of physical activity and sedentary behavior using un objective method. A weakness of the work is that the research design does not allow to talk about cause-effect relationships, so it is suggested that the authors do not talk about the effects of physical activity on psychological distress, but about relationship between physical activity and psychological distress.
The introduction is unconvincing, and it gives an incomplete view of the state-of-the-art. In the second paragraph, it is mentioned that psychological problems, including dementia and depression, have become a public health challenge in Japan, but the relationship between dementia and psychological distress, which is the psychological variable studied in the manuscript, is not mentioned.
In the abstract, in lines 19 and 20 it is “Sleeping time was evaluated using a self-reported questionnaire as well as clinical parameters” but in the rest of the manuscript the clinical parameters evaluated are not described, and in section 2.2. Clinical Parameters, in lines 67 and 68 it says “Total sleeping time (minutes/day) and time in bed (minutes/day) were evaluated by self-reported questionnaire for seven consecutive days”.
The information on the multiple linear regression used and the results obtained in said regression should be expanded, since in the results and discussion section it is alluded to that walking time (minutes/day) and total sleeping time (minutes/day) were important factors for the K6 scores even after adjusting for age, sex and sedentary behavior, but the results of the regression that are given in the text and in Table 3 are not enough to sustain such affirmation.
Round 2
Reviewer 1 Report
In general, the authors have taken my previous comments into consideration. However, I think some of my previous notes were not correctly interpreted. I have further suggestions regarding the manuscript.
Title
§ To promote consistency throughout the text, I’d suggest that the title is “Relationship among Physical Activity, Sleeping Time and Psychological Distress in Community-Dwelling Elderly Japanese”.
Introduction
§ In line 35, I find the prevalence to be too low, considering data on papers such as “Depression in Japanese community-dwelling elderly--prevalence and association with ADL and QOL.” by Wada et al. (2004) and “Community social capital and inequality in depressive symptoms among older Japanese adults: A multilevel study” by Haseda et al. (2018).
§ In lines 39-40, I find the justification for not exploring the relationship between smoking and drinking habits and psychological distress to be weak. Despite the prevalence of such habits, it is possible that there is a relationship between the variables. Considering that previous studies support such a relationship and that these data were collected for the study, I believe smoking and drinking habits should at least be controlled for in the regression analysis.
§ In lines 50-51, my comment on the previous version is still applicable: is this sample representative of Japanese elders, considering that these participants were community-dwelling and lived in a rural area? If, in fact, representativeness is threatened, this should be highlighted in the study limitations.
Materials and methods
§ In lines 54-58, a rationale for excluding participants with injury and/or illness should be provided.
§ I believe the authors did not understand my comment on the previous version regarding the assessment moment. In lines 56-57, the authors state that the participants were enrolled in a health promotion class. At the time of the study, the participants had already participated in that class or the assessment was previous to the class?
Discussion
· In lines 120-121, the authors should clearly state that the variables were associated.
· In lines 145-151, some information is redundant (e.g., regarding the negative relationship between sleeping time and psychological distress).
